# Negative cell cycle regulation by calcineurin is necessary for proper beta cell regeneration in zebrafish

Laura Massoz[1]*, David Bergemann[1], Arnaud Lavergne[1,2], Célia Reynders[1], Caroline Désiront[1], Chiara Goossens[1], Lydie Flasse[1], Bernard Peers[1], Marianne M Voz[1], Isabelle Manfroid[1]

[1]Zebrafish Development and Disease Models Laboratory, GIGA-Stem Cells, University of Liège, Liège, Belgium; [2]GIGA-Genomics Core Facility, GIGA, University of Lièg, Liège, Belgium

**Abstract** Stimulation of pancreatic beta cell regeneration could be a therapeutic lead to treat diabetes. Unlike humans, the zebrafish can efficiently regenerate beta cells, notably from ductal pancreatic progenitors. To gain insight into the molecular pathways involved in this process, we established the transcriptomic profile of the ductal cells after beta cell ablation in the adult zebrafish. These data highlighted the protein phosphatase calcineurin (CaN) as a new potential modulator of beta cell regeneration. We showed that CaN overexpression abolished the regenerative response, leading to glycemia dysregulation. On the opposite, CaN inhibition increased ductal cell proliferation and subsequent beta cell regeneration. Interestingly, the enhanced proliferation of the progenitors was paradoxically coupled with their exhaustion. This suggests that the proliferating progenitors are next entering in differentiation. CaN appears as a guardian which prevents an excessive progenitor proliferation to preserve the pool of progenitors. Altogether, our findings reveal CaN as a key player in the balance between proliferation and differentiation to enable a proper beta cell regeneration.

*For correspondence: laura.massoz@doct.uliege.be

Competing interest: The authors declare that no competing interests exist.

## eLife assessment

This work presents some **valuable** information regarding the molecular mechanisms controlling the regeneration of pancreatic beta cells following induced cell ablation in zebrafish. Specifically, the data suggest that Calcineurin is a regulator of beta cell regeneration. However, the study lacks the critical lineage tracing results to support the conclusion about the origin of the regenerated beta cells and thus is deemed **incomplete**.

## Introduction

Blood glucose homeostasis is tightly controlled by pancreatic endocrine cells. Insulin-producing beta cells work in close association with alpha cells, which secrete glucagon, and delta cells secreting somatostatin, to maintain normal glycemia. The insulin plays a critical role in this process as it is the only hormone able to lower the glycemia. Beta cell loss is a hallmark of type 1 and of late stages of type 2 diabetes, leading to chronic hyperglycemia. Beta cell destruction is largely irreversible in human and in mammals, making the disease incurable nowadays. Nevertheless, studies in diabetic mice models uncovered promising evidences of slight recovery of beta cells via regeneration. For example, new beta cells can arise from the replication of remaining beta cells (*Dor et al., 2004*). Different models of pancreas injuries revealed the plasticity of mammalian pancreatic cells. Actually, differentiated

endocrine cells such as alpha cells (*Thorel, 2010*) and delta cells (*Chera, 2014*) can reprogram and convert into insulin-producing cells, a process that is age dependent. Another possible source of beta cells could be de novo formation from pancreatic progenitors residing in the ductal compartment of the adult pancreas (*Xu et al., 2008*). Lineage tracing experiments could not clearly highlight new beta cells arising from the ductal cells in the adult (*Xu et al., 2008*; *Solar et al., 2009*; *Zhao et al., 2021*). However, expression of the pro-endocrine marker Neurog3 was detected in the ducts in different mouse models of pancreas regeneration (*Xu et al., 2008*; *Kopp et al., 2011*; *Courtney et al., 2013*). In addition, a rare population of ductal cells expressing Neurog3 has been reported to contribute to beta cell neogenesis during diabetes (*Gribben et al., 2021*). A recent study of single-cell RNA sequencing reveal a sub-population of human ductal cells which are able to give rise to all the pancreatic cell types, including beta cells, following implantation in mice (*Qadir et al., 2020*). Altogether, these findings suggest that mammalian pancreatic ducts possess the intrinsic capacity to (re)generate beta cells even though this process is poorly efficient and slow, especially in adults. In contrast to mammals, zebrafish possess remarkable capacity of regeneration, independently of its age (*Moss et al., 2009*; *Poss et al., 2002*; *Choi et al., 2014*; *Goldshmit et al., 2012*; *Kroehne et al., 2011*; *Gemberling et al., 2013*). This model can therefore be exploited to identify and characterize regenerative mechanisms and ultimately induce regeneration in mammals. Based on regenerative mechanisms identified in zebrafish, several studies succeed to improve regeneration in mammals, underlying the possibility of translation from zebrafish to mammals (*Papadimitriou et al., 2018*; *Mashkaryan et al., 2020*; *Ko et al., 2016*; *Goldshmit et al., 2015*; *Fausett et al., 2008*; *Massoz et al., 2021*, for review).

In zebrafish, the nitroreductase (NTR)/nitroaromatic prodrug system is widely used. In this technique, the combination of cell type-specific NTR expression and nitroaromatic prodrug exposure allow for controlled and targeted cell ablation (*Pisharath, 2007*; *Curado et al., 2007*). When NTR is expressed under the control of the *insulin* (*ins*) promoter (*Tg(ins:NTR-mCherry)*; *Pisharath et al., 2007*), beta cell destruction is complete within 3 days following the treatment in adult fish, which correlate with a peak of hyperglycemia (*Delaspre et al., 2015*; *Ghaye et al., 2015*). The regeneration of beta cells upon ablation is spontaneous and fast and the glycemia is normalized within 2 weeks (*Choi et al., 2014*; *Ye et al., 2015*). Similar to mice, new beta cells can arise through proliferation of surviving beta cells (*Andersson, 2012*) as well as through the contribution of alpha cells (*Ye et al., 2015*; *Helker et al., 2019*) and delta cells (*Pardo, 2022*; *Singh et al., 2022*), underscoring the overall conservation of these processes from zebrafish to mammals. Nevertheless, unlike mammals, the presence of pancreatic progenitors in the ducts is well established in both larval (*Ninov et al., 2013*) and adult zebrafish (*Delaspre et al., 2015*; *Ghaye et al., 2015*). Lineage tracing experiments pointed out that ductal cells and centroacinar cells (CACs) give rise to new beta cells (*Delaspre et al., 2015*; *Ninov et al., 2013*). In adults, duct-derived beta cells start to be detected between 7 and 10 days following beta cell ablation (*Delaspre et al., 2015*; *Ghaye et al., 2015*).

Notch signaling is a key player regulating the differentiation of duct-associated progenitors into endocrine cells. Larval duct cells as well as adult CACs display strong Notch activity (*Delaspre et al., 2015*; *Parsons et al., 2009*). This signaling pathway has a central role in beta cell genesis during both development (*Ninov et al., 2012*) and regeneration (*Ninov et al., 2013*) by repressing endocrine differentiation. In zebrafish, different levels of Notch activity determine the behavior of the pancreatic progenitors. While a high level of Notch activity maintains cells in quiescence, a moderate level induces the entry in the cell cycle and proliferation whereas a low level drives endocrine differentiation of the progenitor cells (*Ninov et al., 2012*). A steep decrease of Notch activity pushes the progenitors to differentiate prematurely, bypassing the amplification step and leading to their depletion (*Ninov et al., 2012*). Repression of the Notch signaling by mTor, activated by glucose and nutrients, hence promotes beta cell formation and regeneration from ductal progenitors (*Ninov et al., 2013*). While Notch and mTor signaling are crucial for this process, there is still a need to establish a global view of the molecular mechanisms regulating beta cell regeneration.

To identify early events regulating ductal-derived beta cell regeneration, we determined the transcriptomic signature of ductal cells from adult zebrafish following beta cell destruction. Our data highlighted an upregulation of the calcineurin (CaN) pathway. To elucidate CaN function in beta cell regeneration, we both repressed and activated CaN pathway. We showed that CaN regulates beta cell neogenesis in the ducts during regeneration, by acting on progenitor proliferation. Together, our

findings underline that CaN fine tunes the balance between progenitor proliferation and beta cell differentiation to guarantee proper regeneration.

## Results

### Transcriptomic profiling of ductal cells after beta cell destruction highlights regulation of CaN pathway

To gain a better understanding of the molecular mechanisms underlying the regeneration of beta cells from the ducts, we determined the transcriptional landscape of ductal cells by RNA-sequencing after beta cell ablation in the adult zebrafish. To selectively ablate the beta cells, we used the *Tg(ins:NTR-mCherry)* transgenic fish. The ductal cells were labeled thanks to the *Tg(nkx6.1:GFP)* reporter line (*Ghaye et al., 2015*) in which Green Fluorescent Protein (GFP) marks the ductal tree and associated multipotent pancreatic progenitors (*Ghaye et al., 2015*). More precisely, three to four fishes were treated with the prodrug MTZ at 10 mM overnight, to induce ablation of beta cells or with DMSO(Dimethyl sulfoxide) for the non-ablated controls. Ablation was confirmed by blood glucose measurement before collection of the pancreas. To capture the early events triggered by the destruction of beta cells, we generated the transcriptome of the ducts 3 days post ablation treatment (dpt), that is before beta cell neogenesis (*Figure 1—figure supplement 1*). Differential gene expression analysis revealed that 1866 genes are upregulated and 1515 genes downregulated in the ductal cell of fish treated with MTZ compared to control ($p_{adj} < 0.05$). According to Gene Ontology (GO) analysis, the most enriched pathways among the upregulated genes were DNA replication and cell cycle (*Figure 1A*). This further corroborates our previous findings and those of others, regarding the activation of duct-associated progenitors' proliferation in response to beta cell ablation (*Delaspre et al., 2015*; *Ghaye et al., 2015*). As expected, the Notch pathway was enriched in the downregulated genes (*Ninov et al., 2013*; *Ninov et al., 2012*; *Figure 1B*). On a contrasting note, in a context marked by a robust proliferative response, we found it intriguing to observe an enrichment in the cellular senescence signature (*Figure 1A*). Subsequently, we investigated deeper into the genes associated with this specific signature. We found among them several components of the CaN signaling pathway such as *nfatc3b*, *ppp3ccb* (the catalytic subunit of CaN), *itpr2*, and *calm3b* (*Figure 1C*). In addition to these genes related to the cellular senescence signature, our transcriptomic studies revealed the modulation of other genes from the CaN canonical pathway (*Figure 1D*), underlying its potential role in beta cell regeneration. CaN is a highly conserved calcium/calmodulin-dependent Ser/Thr phosphatase, involved in numerous biological process including fin regeneration and beta cell function (*Tornini, 2016*; *McMillan et al., 2018*; *Cao et al., 2021*; *Kujawski et al., 2014*). This prompted us to investigate the role of CaN in beta cell regeneration.

Previous RNAseq data performed in our laboratory indicate that CaN (*ppp3cca/b*) and NFATc3 (*nfatc3a/b*) are mainly express in endocrine cells (*Tarifeño-Saldivia et al., 2017*; *Figure 1—figure supplement 1(2)*), which is in accordance with the role of CaN/NFAT signaling in beta cells (*Heit et al., 2006*). CaN genes (*ppp3cca/b*) as well as *nfatc3b* are express at lower levels in the ducts at basal state but their expression is induced in response to beta cell destruction (*Figure 1—figure supplement 1(3)*).

### CaN activity regulates the ductal regenerative response

To evaluate the role of CaN in beta cell regeneration, more specifically derived from ductal progenitors, we have chosen to use young larvae, where regenerated beta cells in the pancreatic tail arise exclusively from the ducts (*Ninov et al., 2013*). In response to beta cells ablation, the intrapancreatic ducts undergo a ductal regenerative response whereby differentiation toward the endocrine fate is increased (*Ninov et al., 2013*). We first determined the rate of beta cell neogenesis from the ducts in response to a single acute ablation of beta cells as we performed in adults. We treated *Tg(ins:NTR-P2A-mCherry); Tg(nkx6.1:GFP)* larvae with nifurpirinol (NFP) from 3 to 4 dpf and mCherry+ beta cells were quantified in the GFP+ ducts in the pancreatic tail at several time points: 4, 7, 10, and 14 dpt (*Figure 1E*). Duct-associated beta cells started to be detected in non-ablated larvae between 7 and 10 dpt (*Figure 1F*) and the number of beta cells slowly increased until 14 dpt (*Figure 1F, G*). In ablated larvae, the increase became more pronounced from 10 dpt onwards (*Figure 1F, G*), indicating faster endocrine differentiation. This experiment establishes that the ductal regenerative response is

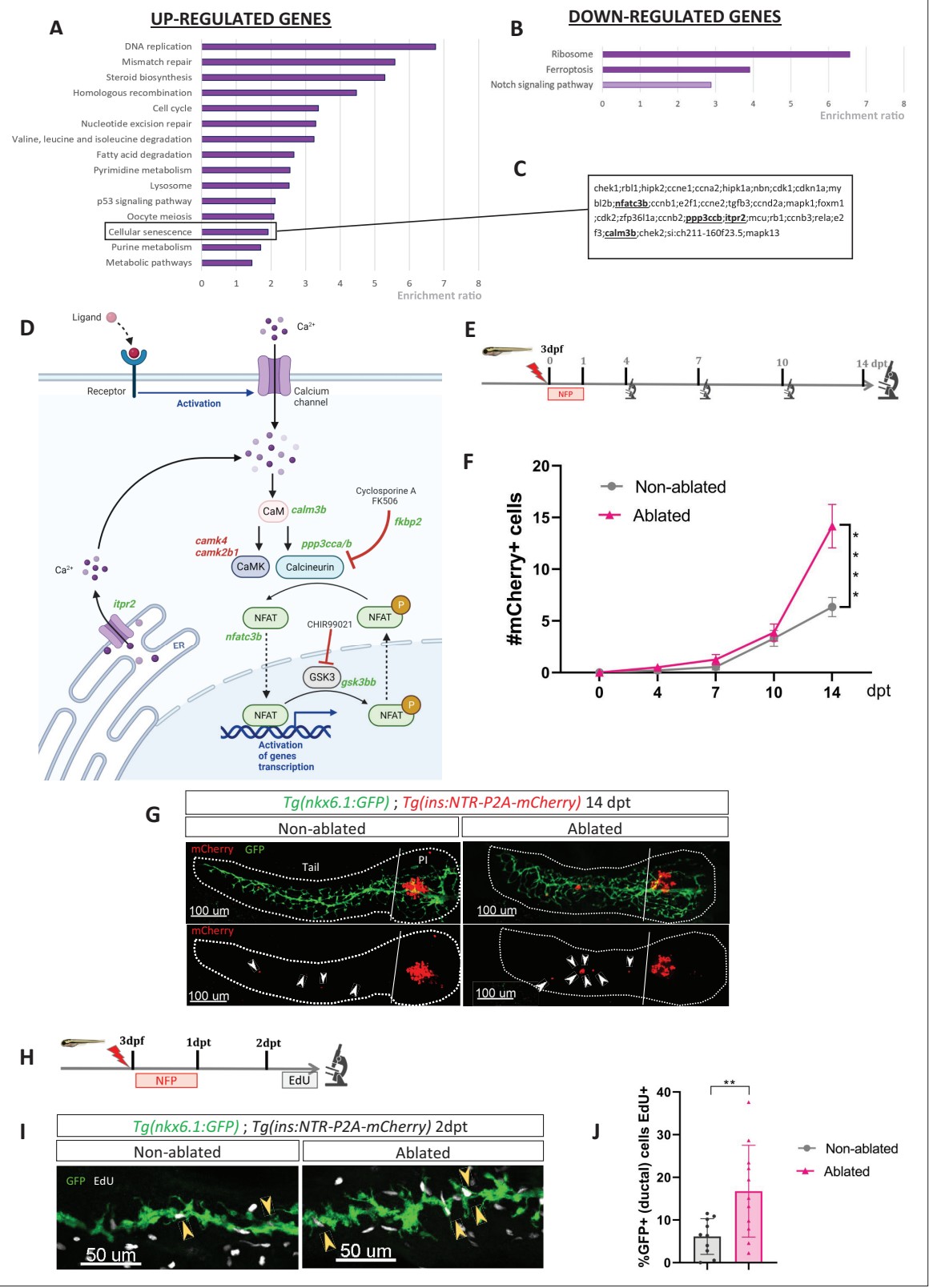

**Figure 1.** Transcriptomic profiling of ductal cells during beta cell regeneration and validation in larvae. (**A, B**) Enrichment ratio of selected non-redundant signatures of Kyoto Encyclopedia of Genes and Genomes (KEGG) pathways overrepresented in ductal cells after beta cells ablation (UP – A and DOWN – B) compared to ductal cells without beta cells ablation. Gene Ontology (GO) terms were identified using over-representation analysis (ORA) analysis by WebGestalt using the list of differentially expressed (DE) genes provided by DESeq. The light color for Notch pathway means p-value

*Figure 1 continued on next page*

*Figure 1 continued*

= 0.11. (**C**) List of genes associated with the signature of cellular senescence from A and B. Genes related to calcineurin (CaN) pathway are in bold. (**D**) CaN canonical pathway with upregulated genes in green and downregulated genes in red in transcriptomic data from **A and B**. (**E**) Experimental design for regeneration test in larvae. Briefly, after nifurpirinol (NFP) treatment from 3 to 4 dpf, larvae were fixed and analyzed at 4–7–10 and 14 days post treatment (dpt). (**F**) Graph representing the mean number of mCherry+ beta cells in the pancreatic tail of *Tg(ins:NTR-P2A-mCherry); Tg(nkx6.1:GFP)* at 0–4–7–10 and 14 dpt. The gray spheres represent non-ablated conditions and the pink triangles the ablated condition. Data are presented as mean values ± SEM. One-way ANOVA test with Tukey's multiple comparison test, ****p-value <0.0005. The experiment was performed at least two times and the data were combined in this graph. (**G**) Whole mount fluorescent immunohistochemistry (GFP and mCherry) of the pancreas of *Tg(ins:NTR-P2A-mCherry); Tg(nkx6.1:GFP)* larvae at 14 dpt. 3D projection (stack) of one non-ablated and one ablated representative samples. The principal islet (PI) and the pancreatic tail are shown. Arrows point out mCherry+ beta cells in the pancreatic tail. Scale 100 μM. (**H**) Experimental design for 5-ethynyl-2'-deoxyuridine (EdU) assay in larvae. After NFP treatment for 3 to 4 dpf, larvae were exposed to EdU at 2 dpt before fixation for analysis. (**I**) Whole mount fluorescent immunohistochemistry (GFP and EdU) of the pancreatic tail of *Tg(ins:NTR-P2A-mCherry); Tg(nkx6.1:GFP)* larvae at 2 dpt. 3D projection (stack) of one non-ablated and one ablated representative samples. Arrows point out GFP+ duct cells EdU+ in the pancreatic tail. Scale 50 μM. (**J**) Barplot representing the percentage of GFP+ ductal cells which incorporated EdU+ in non-ablated (n=10) and ablated conditions (n=11). Data are presented as mean values ± standard deviation (SD). *T*-test. **p-value <0.005. The experiment was performed at least two times.

The online version of this article includes the following figure supplement(s) for figure 1:

**Figure supplement 1.** Transcriptomic profiling of ductal cells during beta cell regeneration and validation in larvae.

detectable between 10 and 14 days after the beta cell ablation, performed at 3 dpf. We next wanted to determine if ductal cell proliferation is activated in response to beta cell destruction in larvae as in adult fish. We exposed *Tg(ins:NTR-P2A-mCherry); Tg(nkx6.1:GFP)* larvae to 5-ethynyl-2'-deoxyuridine (EdU) the second day following ablation (*Figure 1H*). In the ablated larvae, the proportion of GFP+ ductal cells EdU+ (in S-phase) was higher compared to non-ablated larvae (*Figure 1I, J*). This result shows that acute beta cell ablation in larvae rapidly activates ductal cell proliferation, as previously reported in adult zebrafish.

As our transcriptomic data from adult zebrafish revealed modulation of the CaN pathway at 3 dpt (*Figure 1D*), we treated *Tg*(*ins:NTR-P2A-mCherry); Tg*(*nkx6.1:GFP)* larvae from 1 to 3 dpt after beta cell ablation, with a CaN inhibitor, the Cyclosporin A (CsA) (*Kujawski et al., 2014*; *Figure 2A*). The number of newly formed beta cells in the tail was monitored from 4 to 14 dpt. CsA-enhanced beta cell formation at 10 dpt (*Figure 2B–D*). However, this effect appears to be transient since no discernible difference was observed between the control and CsA-treated larvae in regeneration at the latest time point (14 dpt), suggesting an acceleration of the regenerative response (*Figure 2B*; *Figure 2— figure supplement 1*). Interestingly, CsA did not affect beta cell differentiation in non-ablated larvae, indicating that CsA only acts in a regenerative context (*Figure 2B–D*). Of note, CsA increased as well the number of regenerated beta cells in the principal islet (*Figure 2—figure supplement 1(2) and (3)*). We and others, previously showed that the othercegeneration, besides the ducts, are the bi-hormonal sst1.1+/ins+ cells (*Pardo, 2022*; *Singh et al., 2022*; *Mi et al., 2023*). However, CsA did not affect bi-hormonal cell formation (*Figure 2—figure supplement 1(4)*), suggesting that the additional cells in the principal islet could also originate from the ducts.

We next tested the effect of CsA on endocrine progenitors in a regenerative context. We induced regeneration in Tg(*neurod1:GFP*) larvae where the GFP is expressed in both endocrine progenitors and mature endocrine cells. We first assessed generation of GFP cells at different time points (*Figure 2—figure supplement 1(5)*) and showed that CsA induced an increase of neurod1+ cells from 4 dpt. The increase was still detectable at least until 10 dpt (*Figure 2E*, *Figure 1—figure supplement 1(6)*). To determine if these additional cells result from their own proliferation, we performed a pulse of EdU just before analysis (*Figure 2—figure supplement 1(5)*). We observed that CsA did not affect the neurod1+ cell proliferation rate, which is very low at these stages (*Figure 2F*). As a consequence, the effect of CsA cannot be explained by endocrine cell proliferation but rather by neogenesis from progenitors. As CsA affects pro-endocrine cells formation, we next wondered if the increased cell formation induced by CsA is specific to beta cells. Treatment with CsA was performed as previously and delta1.1 and alpha cells were detected by immunofluorescence. Interestingly, CsA did not affect alpha nor delta1.1 cells neogenesis in response to beta cell ablation (*Figure 2—figure supplement 1(7), (8)*). Overall, these experiments showed that CsA affects specifically the beta cells and their endocrine progenitors.

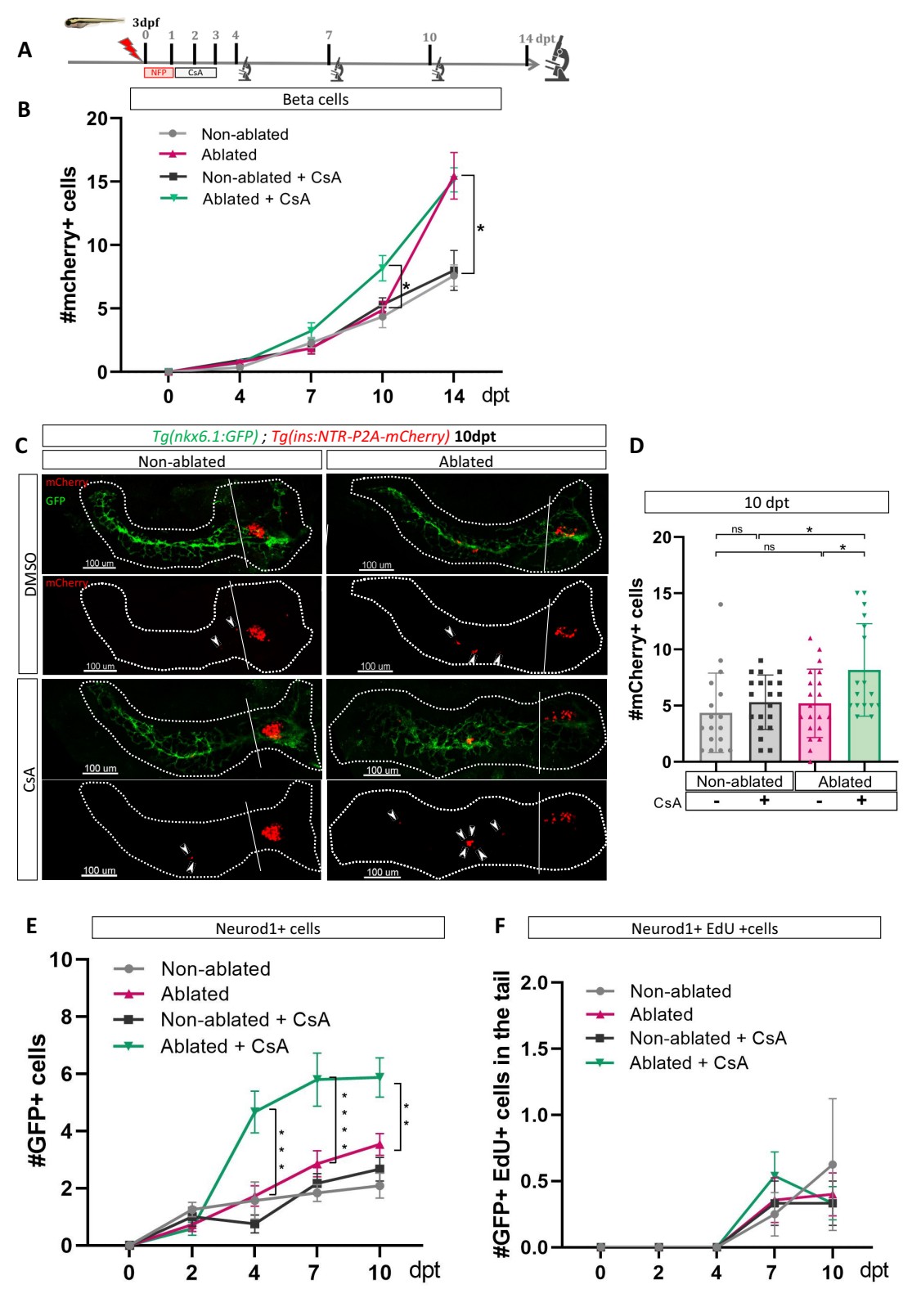

**Figure 2.** Calcineurin inhibition with Cyclosporin A (CsA) increases the ductal regenerative response. (**A**) Experimental design for regeneration test in larvae with CsA treatment. Briefly, after nifurpirinol treatment from 3 to 4 dpf, larvae were treated with CsA from 1 to 3 dpt and fixed and analyzed at 4–7–10 and 14 days post treatment (dpt). The experiment was performed at least two times and the data are combined in these graphs. (**B**) Graph representing the mean number of mCherry+ beta cells in the pancreatic tail of *Tg(ins:NTR-P2A-mCherry); Tg(nkx6.1:GFP)* at 0–4–7–10 and 14 dpt.

*Figure 2 continued on next page*

*Figure 2 continued*

The gray spheres represent non-ablated condition; the pink triangles represent the ablated condition; the black squares CsA condition and inverted green triangles ablated + CsA condition. Data are presented as mean values ± SEM. Two-way ANOVA test with Sidak's multiple comparisons test, *p-value <0.05. (**C**) Whole mount fluorescent immunohistochemistry (GFP and mCherry) of the pancreas of *Tg(ins:NTR-P2A-mCherry); Tg(nkx6:GFP)* larvae at 10 dpt. 3D projection (stack) of non-ablated and ablated larvae treated with DMSO or CsA representative samples. The principal islet (PI) and the pancreatic tail are shown. Arrows point out mCherry+ beta cells in the pancreatic tail. Scale 100 μM. (**D**) Barplot representing the number of number of mCherry+ beta cells in the pancreatic tail of *Tg(ins:NTR-P2A-mCherry); Tg(nkx6.1:GFP)* larvae at 10 dpt. The gray spheres represent non-ablated condition (n=17) ; the pink triangles represent the ablated condition (n=20); the black squares CsA condition (n=20) and inverted green triangles ablated + CsA condition (n=17). Data are presented as mean values ± standard deviation (SD). Two-way ANOVA with Tukey's multiple comparison test, *p-value <0.05. (**E**) Graph representing the mean number of GFP+ neurod1+ cells in the pancreatic tail of *Tg(ins:NTR-P2A-mCherry); Tg(neurod1:GFP)* at 0–4–7 and 10 dpt. The gray spheres represent non-ablated condition; the pink triangles represent the ablated condition; the black squares CsA condition and inverted green triangles ablated + CsA condition. Data are presented as mean values ± SEM. Two-way ANOVA test with Sidak's multiple comparisons test, **p-value <0.005; ***p-value <0.0005; ****p-value <0.00005. The experiment was performed at least two times and the data are combined in the graph. (**F**) Graph representing the mean number of GFP+ neurod1 EdU+ cells in the pancreatic tail of *Tg(ins:NTR-P2A-mCherry); Tg(neurod1:GFP)* at 0–4–7 and 10 dpt. The gray spheres represent non-ablated condition; the pink triangles represent the ablated condition; the black squares CsA condition and inverted green triangles ablated + CsA condition. Data are presented as mean values ± SEM.

The online version of this article includes the following figure supplement(s) for figure 2:

**Figure supplement 1.** Calcineurin inhibition with Cyclosporin A (CsA) increases the ductal regenerative response.

## CaN over-activation abolishes the regenerative response

We then wondered if an opposite regulation of CaN that is its activation impacts as well the regenerative response. To that end, we generated a transgenic line Tg(*hsp70:GFP-P2A-ppp3ccCA*) that allows ubiquitous expression of a constitutively active form of CaN$^{CA}$ (*ppp3cc$^{CA}$*) upon heat-shocks. Beta cell ablation was triggered in Tg(*hsp70:GFP-P2A-ppp3cc$^{CA}$*); Tg(*ins:NTR-P2A-mCherry*); Tg(*nkx6.1:GFP*) larvae from 3 to 4 dpf and CaN$^{CA}$ expression was induced by four successive heat-shocks from 1 to 3 dpt (**Figure 3A**). The overexpression of CaN$^{CA}$ after ablation impaired the regenerative response at 14 dpt (**Figure 3B, C**). Similar results were obtained with Tg(*UAS:GFP-P2A-ppp3cc$^{CA}$*); Tg(*cftr:gal4*) (**Liu et al., 2018**); Tg(*ins:NTR-P2A-mCherry*) larvae in which CaN$^{CA}$ is continuously and specifically overexpressed in the ducts within the pancreas (**Figure 3D, E**). Importantly, the structure of the ducts was similar in CaN$^{CA}$ overexpressing larvae compared to Tg(*nkx6.1:GFP*) controls (**Figure 3D**), suggesting that the suppression of the regenerative response in CaN$^{CA}$-overexpressing larvae was not due to morphogenetic defects during ductal growth. This result shows that CaN acts directly in the ducts to decrease beta cell regeneration while it is not necessary for normal beta cell differentiation.

## CaN regulates beta cell differentiation induced by Notch inhibition in absence of regeneration

Our transcriptomic data showed that the Notch pathway is downregulated in ductal cells during beta cell regeneration (**Figure 1**). As the level of Notch activity determines the behavior of ductal cells (**Ninov et al., 2013**) from quiescence to proliferation and subsequently to beta cell differentiation, we tested whether CaN acts together with the Notch pathway on a common pool of ductal progenitors. To inhibit the Notch pathway, we treated larvae with several concentrations of the gamma-secretase inhibitor LY411575 from 3 to 4 dpf in absence of regeneration. The activity of CaN was inhibited by CsA during the same timeframe (**Figure 4A**). As previously, we used reporter lines for beta and ductal cells *Tg(ins:NTR-P2A-mCherry); Tg(nkx6.1:GFP)* and the number of secondary beta cells was analyzed at 6 dpf (**Figure 4A**). As expected, the number of beta cells progressively rose as the concentration of the Notch inhibitor increased (**Figure 4B**). Combined treatment with CsA fostered the differentiation of beta cells between 1 and 10 μM LY411575 but did not result in further increase at 15 μM LY411575 (**Figure 4B**), suggesting that CaN is important within a permissive window of Notch activity. Since we overserved the highest synergistic effect at 5 μM of LY411575, we used this concentration for the following experiments (**Figure 4C, D**). It is worth noting that combined treatment of LY411575 (5 μM) and another CaN inhibitor, FK506, resulted in the same synergistic increase of beta cell differentiation (**Figure 4—figure supplement 1(1), (2)**), confirming that the effect is well due to CaN inhibition. The combined effect of Notch and CaN inhibition is transient as it is not observed at 7 dpf anymore (**Figure 4E**). Therefore, as observed in regenerative conditions (**Figure 2B**), CaN inhibition accelerates beta cell neogenesis induced by Notch repression.

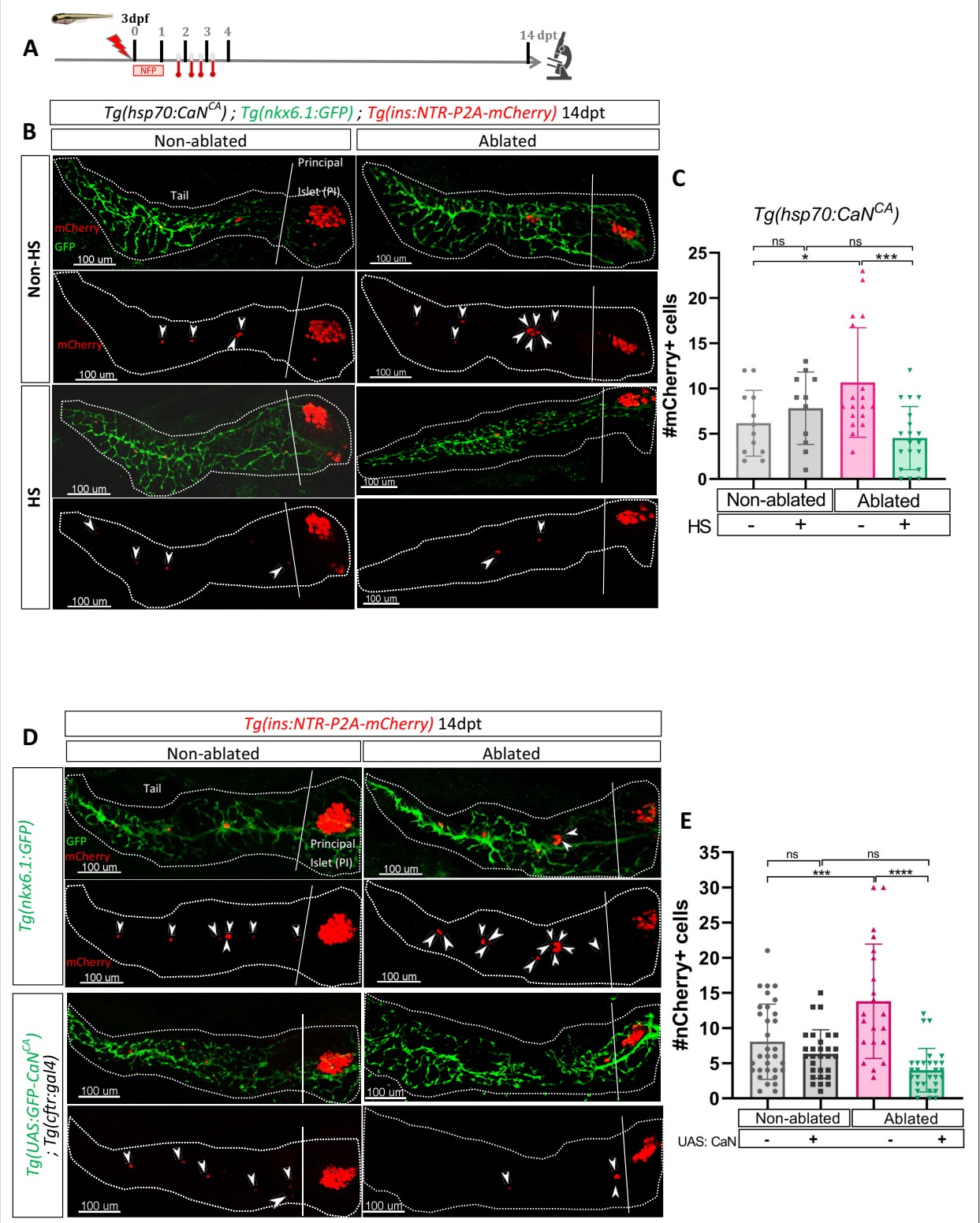

**Figure 3.** Transgenic mediated overexpression of calcineurin abolish the ductal regenerative response. (**A**) Experimental design for regeneration test in larvae with heat-shocks. Briefly, after nifurpirinol treatment from 3 to 4 dpf, four heat-shock were performed from 1 to 3 dpt and larvae were fixed and analyzed at 14 dpt. (**B**) Whole mount fluorescent immunohistochemistry (GFP and mCherry) of the pancreas of *Tg(hsp70:CaN^CA); Tg(ins:NTR-P2A-mCherry); Tg(nkx6.1:GFP)* larvae at 14 dpt. 3D projection (stack) of one non-ablated and one ablated with or without heat-shock representative samples. The principal islet (PI) and the pancreatic tail are showed. Arrows point out mCherry+ beta cells in the pancreatic tail. Scale 100 μM. (**C**) Barplot

*Figure 3 continued on next page*

*Figure 3 continued*

representing the mean number of mCherry+ beta cells in the pancreatic tail of *Tg(hsp70:CaN^CA)*; *Tg(ins:NTR-P2A-mCherry)*; *Tg(nkx6.1:GFP)* larvae at 14 dpt. The gray spheres represent non-ablated condition (n=12); the pink triangles represent the ablated condition (n=18); the black squares heat-shock condition (n=11) and inverted green triangles ablated + heat-shock condition (n=19). Data are presented as mean values ± standard deviation (SD). Two-way ANOVA with Tukey's multiple comparisons test, *p-value <0.05, ***p-value <0.0005, ns = non-significant. The experiment was performed at least two times and the data are combined in the graph. (**D**) Whole mount fluorescent immunohistochemistry (GFP and mCherry) of the pancreas of larvae at 14 dpt. 3D projection (stack) of *Tg(ins:NTR-P2A-mCherry)*; *Tg(nkx6.1:GFP)* one non-ablated and one ablated representative control samples and *Tg(UAS:CaN^CA)*; *Tg(ins:NTR-P2A-mCherry)*; *Tg(cftr:gal4)* one non-ablated and one ablated representative samples. The principal islet (PI) and the pancreatic tail are showed. Arrows point out mCherry+ beta cells in the pancreatic tail. Scale 100 μM. (**E**) Barplot representing the mean number of mCherry+ beta cells in the pancreatic tail of larvae at 14 dpt. The gray spheres represent non-ablated *Tg(ins:NTR-P2A-mCherry)*; *Tg(nkx6.1:GFP)* condition (n=31); the pink triangles represent the ablated *Tg(ins:NTR-P2A-mCherry)*; *Tg(nkx6.1:GFP)* condition (n=21); the black squares non-ablated *Tg(UAS:CaN^CA)*; *Tg(ins:NTR-P2A-mCherry)*; *Tg(cftr:gal4)* condition (n=29) and inverted green triangles ablated *Tg(UAS:CaN^CA)*; *Tg(ins:NTR-P2A-mCherry)*; *Tg(cftr:gal4)* condition. Data are presented as mean values ± SD. Two-way ANOVA with Tukey's multiple comparison test, ***p-value <0.0005, ****p-value <0.00005, ns = non-significant. The experiment was performed at least two times and the data are combined in the graph.

To determine to which extent CaN pathway can modulate Notch-induced beta cell neogenesis, we activated CaN^CA overexpression in *Tg(hsp70:GFP-P2A-ppp3cc^CA)*; *Tg(ins:NTR-P2A-mCherry)* larvae by an heat-shock at 3 dpf and treated them with LY411575 (*Figure 4—figure supplement 1(3)*). CaN^CA overexpression resulted in a lowered beta cell formation induced by Notch inhibition *Figure 4—figure supplement 1(4)*, revealing that CaN activation counterbalanced the effects of Notch inhibition. Using our previous settings of Notch and CaN inhibition, we next wondered if the canonical pathway downstream of CaN was involved in the enhancement of beta cell differentiation. To activate NFAT, a well-known target of CaN, we used CHIR99021 allowing a stabilization of the active form of NFAT (*Figure 1D*). We found that CHIR99021 rescued the effect of CsA (*Figure 4—figure supplement 1(5)*), suggesting that CaN inhibition increases beta cell neogenesis at least partially by the regulation of NFAT. Overall, these results reveal that CaN impacts beta cell formation in pro-endocrinogenic context, such as induced by a low level of Notch activity. Moreover, it suggests that both CaN and Notch pathways act on a common pool of ductal progenitors to govern beta cell neogenesis.

## CaN controls the proliferation of duct-associated progenitors induced by Notch inhibition

Given that we observed the most significant increase in beta cell formation with CsA when Notch activity was mildly repressed (*Figure 4B–D*), and since mild Notch activity has been shown to promote progenitor amplification (*Ninov et al., 2012*), it suggests that CaN acts at this level. To explore this possibility, we exposed briefly *Tg(nkx6.1:GFP)* larvae to EdU after mild Notch (LY411575 5 μM) and CaN inhibition (*Figure 5A*) and analyzed at 4 and 6 dpf. As expected, Notch inhibition increased the proportion of proliferating EdU+ GFP+ ductal cells at 4 dpf (*Figure 5B, C*) while the number of GFP+ ductal cells remained constant (*Figure 5—figure supplement 1*). Furthermore, the amount of ductal progenitors decreased 2 days later (at 6 dpf) (*Figure 5D*), which is concomitant with the increase of beta cell differentiation (*Figure 4C, D*). Interestingly, the combined inhibition with CsA further increased these proportions, while CaN inhibition alone had no effect (*Figure 5B–D*). Of note, we detected beta cells EdU+ at 6 dpf (*Figure 5—figure supplement 1(2)*), underlying that these cells originate from the proliferation of ductal progenitors.

Incidentally, at 4 dpf, while the proliferation is increased (*Figure 5C*), the number of ductal cells remained the same in all conditions (*Figure 5—figure supplement 1(1)*), suggesting that ductal cells have not yet left the cell cycle to differentiate. In comparison, after stronger Notch inhibition (15 μM), the ductal cells are already depleted at 4 dpf (*Figure 5—figure supplement 1(3)*), as they directly differentiate without entering the cell cycle (*Ninov et al., 2012*). In these conditions, CsA could therefore not enhance ductal progenitor proliferation and thus beta cell formation (*Figure 4B*). These results show that CaN and Notch pathways act together on the proliferation of the ductal progenitors to prevent their exhaustion.

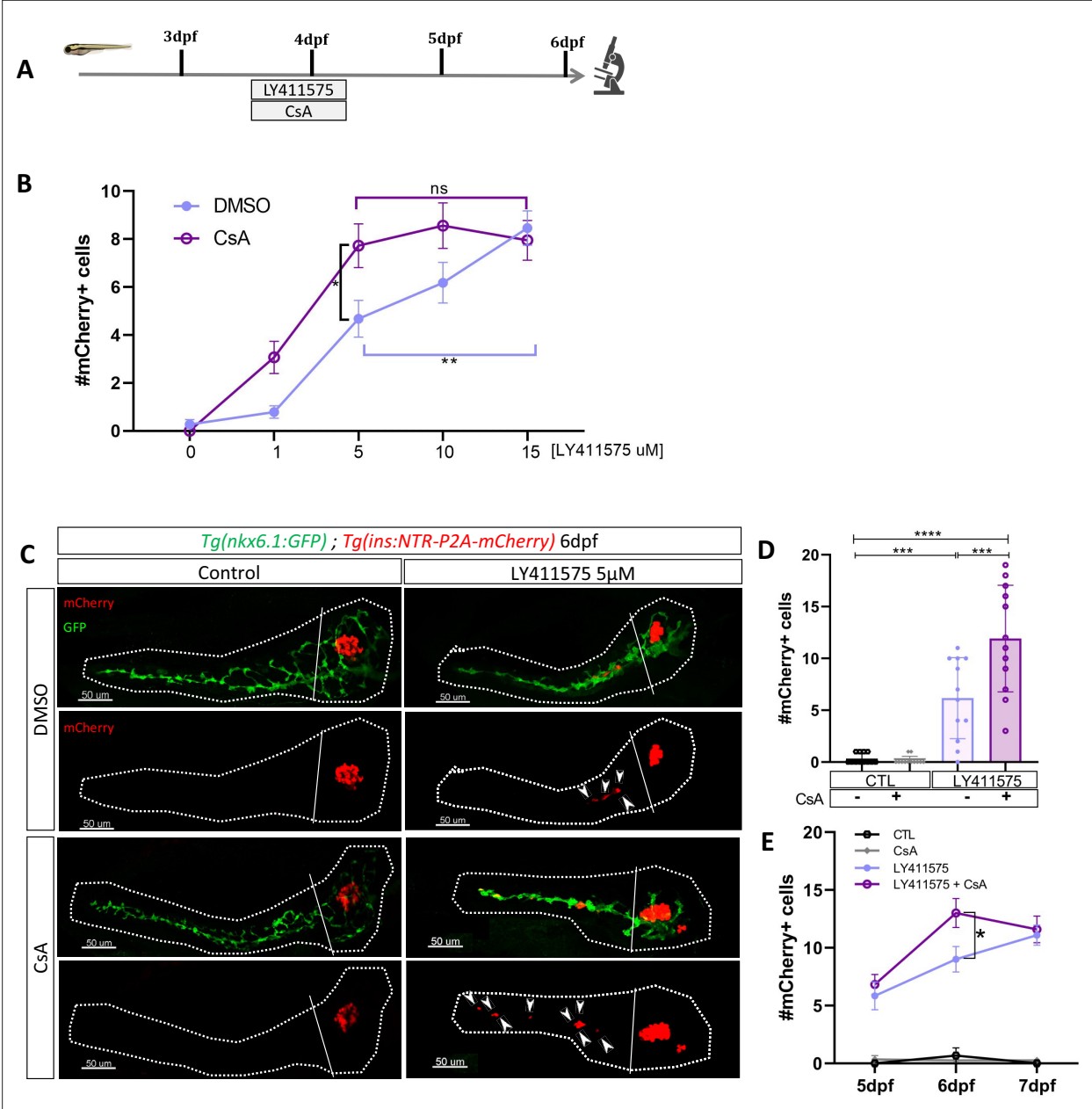

**Figure 4.** Calcineurin (CaN) repression potentializes the effect of Notch inhibition on beta cell formation. (**A**) Experimental design for Notch inhibition test in non-ablated condition. Larvae were treated concomitantly with LY411575 (Notch inhibitor) and Cyclosporin A (CsA) from 3 to 4 dpf and were fixed and analyzed at 6 dpf. (**B**) Graph representing the mean number of mCherry+ beta cells in the pancreatic tail of *Tg(ins:NTR-P2A-mCherry); Tg(nkx6.1:GFP)* larvae at 6 dpf depending the concentration of LY411575. The blue dots represent LY411575; and purple combination of LY411575 and CsA. Data are presented as mean values ± SEM. Two-way ANOVA test with Sidak's multiple comparison test, *p-value <0.05, **p-value <0.05, ns = non-significant. The experiment was performed at least two times and the data are combined in the graph. (**C**) Whole mount fluorescent immunohistochemistry (GFP and mCherry) of the pancreas of *Tg(ins:NTR-P2A-mCherry); Tg(nkx6.1:GFP)* larvae at 6 dpf. 3D projection (stack) of one control (without any treatment); one CsA-treated; one LY411757-treated and one with both CsA- and LY411575-treated larvae. The principal islet (PI) and the pancreatic tail are showed. Arrows point out mCherry+ beta cells in the pancreatic tail. Scale 50 µM. (**D**) Barplot representing the mean number of mCherry+ beta cells in the pancreatic tail of *Tg(ins:NTR-P2A-mCherry); Tg(nkx6.1:GFP)* larvae at 6 dpf. The black dots represent the control (n=12); gray CsA treatment (n=12);blue LY411575 (n=12); and purple combination of LY411575 and CsA (n=12). Data are presented as mean values ± standard deviation (SD). Two-way ANOVA with Tukey multiple comparison test, ***p-value <0.0005, ****p-value <0.00005. The experiment was performed at least two times and the data are combined in the graph. (**E**) Graph representing the mean number of mCherry+ beta cells in the pancreatic tail of *Tg(ins:NTR-P2A-mCherry); Tg(nkx6.1:GFP)* larvae at 5–6–7 dpf. The black dots represent the control; gray CsA treatment;blue LY411575; and purple combination

*Figure 4 continued on next page*

*Figure 4 continued*

of LY411575 and CsA. Data are presented as mean values ± SEM. Two-way ANOVA test with Sidak's multiple comparison test, *p-value <0.05. The experiment was performed at least two times and the data are combined in the graph.

The online version of this article includes the following figure supplement(s) for figure 4:

**Figure supplement 1.** Calcineurin (CaN) repression potentializes the effect of Notch inhibition on beta cell formation.

## CaN prevents exhaustion of Notch responsive progenitors during beta cell regeneration

Taken together, our results indicate that CaN plays a role in the proliferation of ductal progenitors in contexts that are permissive for beta cell differentiation. To demonstrate that CaN acts on ductal progenitor proliferation in a similar manner to Notch inhibition but during regeneration, we exposed *Tg(ins:NTR-P2A-mCherry); Tg(nkx6.1:GFP)* larvae after ablation and CsA treatment (*Figure 5E*). CaN inhibition enhanced the proliferation of ductal cells in ablated larvae (*Figure 5F, G*). To next determine if CaN acts on the Notch-responsive progenitors in beta cell regeneration, we then used *Tg(tp1:VenusPest)* Notch reporter line. At larval stages the vast majority of ductal cells are Notch responsive and the reporter line marks all the progenitors within the ductal tree (*Parsons et al., 2009*). We treated *Tg(ins:NTR-P2A-mCherry); Tg(tp1:VenusPest)* as described above (*Figure 5E*). In ablated-larvae, CsA increased tp1+ ductal cell proliferation (*Figure 5H, I*). Moreover, CsA induced a reduction of tp1+ ductal cells in ablated larvae (*Figure 5H–J*), suggesting an exhaustion of the Notch responsive progenitors, in accordance with premature beta cell differentiation we observed (*Figures 2B–4D*). Lastly, CsA does not affect tp1+ cells in non-ablated larvae showing that CaN inhibition did not directly affect Notch signaling (*Figure 5J*). Those results suggest that CaN fine tunes the balance between proliferation of the progenitors and their differentiation to prevent their exhaustion during beta cell regeneration.

## CaN regulation is functionally relevant in adult zebrafish

To further expand upon our findings and investigate their relevance in a non-developmental context, we next ought to determine whether CaN function is maintained in older zebrafish. We used 2-month-old *Tg(ins:NTR-P2A-mCherry); Tg(nkx6.1:GFP)* juveniles fish to perform beta cell ablation followed by CsA treatments. We analyzed the number of small islets (up to five cells) at 7 and 10 dpt. At 7 dpt, CsA increased the number of small islets in ablated juveniles (*Figure 6A, B*) showing that CaN inhibition enhances beta cell regeneration in juvenile zebrafish, as in larvae. It is noteworthy that, as in larvae, we highlighted an acceleration of beta cell regeneration. The increase number of small islets is indeed transient as it not observe anymore at 10 dpt (*Figure 6C*).

Next, we determined the functional impact of CaN overexpression or inhibition by assessing the glycemia at 7, 10, and 14 after beta cell ablation. Overexpression of CaN$^{CA}$ using either *Tg(hsp70:GFP-P2A-ppp3cc$^{CA}$)* or *Tg(cftr:gal4); Tg(UAS:GFP-P2A-ppp3cc$^{CA}$)*, led to an increased glycemia at both 7 and 10 dpt (*Figure 6D, E*), before recovery at 14 dpt (*Figure 6—figure supplement 1*). This indicates that the overexpression of CaN delayed glycemia recovery induced by beta cell regeneration. However, CaN inhibition did not seem to further improve the glycemia (*Figure 6D, E*) probably because the glycemia was already low at 7 dpt (77 mg/dl on average) compared to non-ablated control (50 mg/dl on average) (*Figure 6D, E*). Altogether, these results show that in adult zebrafish also, CaN regulation is necessary to enable beta cell regeneration and for proper recovery of the glycemia after beta cell loss.

## Discussion

Previous drug and genetic screening using zebrafish larvae enabled the identification of several regulators of beta cell regeneration from different pancreatic cellular sources. For example, adenosine has been shown to stimulate beta cell replication (*Andersson, 2012*), igfbp1a (*Lu et al., 2016*) and TGFb suppression (*Helker et al., 2019*) promote alpha-to-beta cell transdifferentiation. As for cdk5 inhibition and folinic acid/Folr1, they promote beta cell regeneration from the pancreatic ducts (*Karampelias et al., 2021*; *Liu et al., 2018*). Here, to identify novel regulators of beta cell regeneration specifically from pancreatic ducts, we carried out a transcriptomic profiling of duct cells following beta

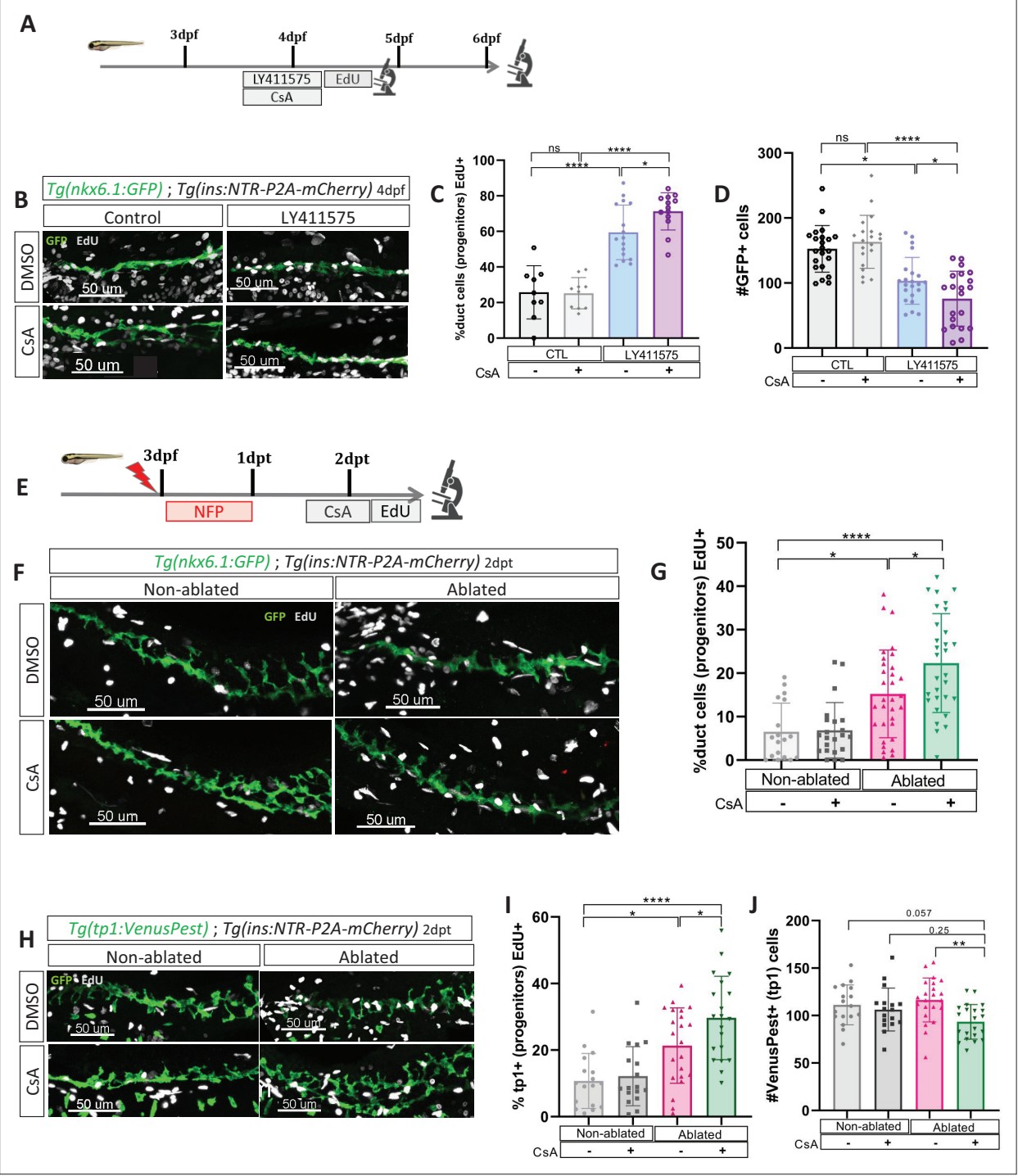

**Figure 5.** Calcineurin (CaN) repression increases the proportion of ductal proliferating cells. (**A**) Experimental design for 5-ethynyl-2'-deoxyuridine (EdU) assay in Notch test. Larvae were treated concomitantly with LY411575 (Notch inhibitor) and Cyclosporin A (CsA) from 3 to 4 dpf and then briefly treated with EdU before fixation and analysis at 4 or 6 dpf. (**B**) Whole mount fluorescent immunohistochemistry (GFP and EdU) of the pancreatic tail of *Tg(ins:NTR-P2A-mCherry); Tg(nkx6.1:GFP)* larvae at 4 dpf. 3D projection (stack) of one control (without any treatment), one with CsA only, one with LY411575 only and one with both CsA and LY411757 representative samples. Scale 50 μM. (**C**) Barplot representing the percentage of GFP+ ductal cells which incorporated EdU+ in pancreatic tail of *Tg(ins:NTR-P2A-mCherry); Tg(nkx6.1:GFP)* larvae for the Notch test. The black dots represent the control (n=9); gray CsA treatment (n=10); blue LY411575 (n=16); and purple combination of LY411575 and CsA (n=16). Data are presented as mean values ± standard deviation (SD). *T*-test. Two-way ANOVA test with Tukey's multiple comparisons test, *p-value <0.05; ****p-value <0.00005; ns = non-significant. The experiment was performed at least two times. (**D**) Barplot representing the number of GFP+ ductal cells which in pancreatic tail of *Tg(ins:NTR-P2A-mCherry); Tg(nkx6.1:GFP)* larvae at 6 dpf for the Notch test. The black dots represent the control (n=22); gray CsA treatment (n=20); blue LY411575

*Figure 5 continued on next page*

*Figure 5 continued*

(n=21); and purple combination of LY411575 and CsA (n=21). Data are presented as mean values ± SD. *T*-test. Two-way ANOVA test with Tukey's multiple comparisons test, *p-value <0.05; ****p-value <0.00005; ns = non-significant. The experiment was performed at least two times and the data are combined in the graph. (**E**) Experimental design for EdU assay in regeneration. Larvae were treated with nifurpirinol for beta cell ablation from 3 to 4 dpf then with CsA from 4 to 5 dpf and then briefly treated with EdU before fixation and analysis. (**F**) Whole mount fluorescent immunohistochemistry (GFP and EdU) of the pancreatic tail of *Tg(ins:NTR-P2A-mCherry); Tg(nkx6.1:GFP)* larvae at 5 dpf. 3D projection (stack) of one representative sample of non-ablated or ablated with or without CsA are shown. Scale 50 µM. (**G**) Barplot representing the percentage of GFP+ ductal cells which incorporated EdU+ in pancreatic tail of *Tg(ins:NTR-P2A-mCherry); Tg(nkx6.1:GFP)* larvae at 5 dpf. The gray spheres represent non-ablated condition (n=18); the pink triangles the ablated condition (n=32); the black squares CsA condition (n=22) and inverted green triangles ablated + CsA condition (n=30). Data are presented as mean values ± SD. Two-way ANOVA test with Tukey's multiple comparisons test, *p-value <0.05; ****p-value <0.00005; ns = non-significant. The experiment was performed at least two times and the data are combined in the graph. (**H**) Whole mount fluorescent immunohistochemistry (VenusPest and EdU) of the pancreatic tail of *Tg(ins:NTR-P2A-mCherry); Tg(tp1:VenusPest)* larvae at 5 dpf. 3D projection (stack) of one representative sample of non-ablated or ablated with or without CsA are shown. Scale 50 µM. (**I**) Barplot representing the percentage of GFP+ ductal cells which incorporated EdU+ in pancreatic tail of *Tg(ins:NTR-P2A-mCherry); Tg(tp1:VenusPest)* larvae at 5 dpf. The gray spheres represent non-ablated condition (n=17); the pink triangles the ablated condition (n=22); the black squares CsA condition (n=18) and inverted green triangles ablated + CsA condition (n=20). Data are presented as mean values ± SD. Two-way ANOVA test with Tukey multiple comparisons test, *p-value <0.05; ****p-value <0.00005; ns means non-significant. The experiment was performed at least two times and the data are combined in the graph. (**J**) Barplot representing the number of VenusPest+ ductal cells which incorporated EdU+ in pancreatic tail of *Tg(ins:NTR-P2A-mCherry); Tg(tp1:VenusPest)* larvae at 5 dpf. The gray spheres represent non-ablated condition (n=17); the pink triangles the ablated condition (n=21); the black squares CsA condition (n=17) and inverted green triangles ablated + CsA condition (n=21). Data are presented as mean values ± SD. Two-way ANOVA test with Tukey's multiple comparisons test, **p-value <0.005. The experiment was performed at least two times and the data are combined in the graph.

The online version of this article includes the following figure supplement(s) for figure 5:

**Figure supplement 1.** Calcineurin (CaN) repression increases the proportion of duct proliferating cells.

cell ablation in the adult zebrafish. Transcriptomic analyses show that the regulated genes encompass most of the genes and pathways identified in previous studies (igfbp1, mTor, Notch, etc.), underlying the importance of those actors in beta cell regeneration. Our data reveal also that DNA replication is the most enriched signature attesting that duct cells undergo a potent proliferative response after the destruction of beta cells.

Besides these expected signatures, our transcriptomic data uncover the unanticipated upregulation of numerous genes implicated in DNA repair and cell cycle arrest. These signatures might indicate that highly proliferating ductal cells activate counteracting mechanisms. Among the genes regulated in those signatures, we focused on CaN and determined its role in beta cell regeneration. In an experimental setting in the young larvae revealing regeneration from the ducts (*Ninov et al., 2013*), pharmacological inhibition of CaN increases the proliferation of duct cells induced by beta cell ablation, resulting in an acceleration of beta cell regeneration in the ducts. Consistently, we also used a genetic approach, and showed that transgene-mediated CaN overactivation abolishes the regenerative response. Importantly, the inhibition of regeneration is observed when CaN is overexpressed either ubiquitously or selectively in *cftr*-expressing ductal cells indicating that the role of CaN in beta cell regeneration is intrinsic to the ducts. Moreover, we showed that CaN acts on the same pool of ductal progenitors than Notch pathway and together control their proliferation and differentiation to beta cells. Altogether, these experiments confirm that the increased of beta cells induced by CaN originate from the ducts in the pancreatic tail. Nonetheless, while these experiments provide strong evidences, performing lineage tracing would further reinforce these data. We also suggest that CaN could improve regeneration in the principal islet, however as the molecular mechanisms of regeneration from the extra pancreatic ducts are not Notch dependant, it would be interesting to investigate the mechanisms under CaN inhibition.

Based on functional assays in larvae, we not only confirm the activation of the proliferation of ductal cells soon after beta cell ablation, but also that the rate of progenitor proliferation is carefully controlled by CaN in order to achieve proper and timely regeneration of beta cells. Our data are consistent with earlier studies reporting a role of CaN in proliferation dynamics during fin regeneration. In the regenerating fin, low CaN activity is found in the proximal region of the blastema characterized by a high rate of proliferation and regeneration and its activity increases distally where lower proliferation is observed (*Tornini, 2016*; *Cao et al., 2021*; *Kujawski et al., 2014*). It was suggested that CaN control blastemal cell progeny divisions (*Tornini, 2016*). In human, the importance of CaN in proliferation is also highlighted in organ transplanted patients. When patients are treated with

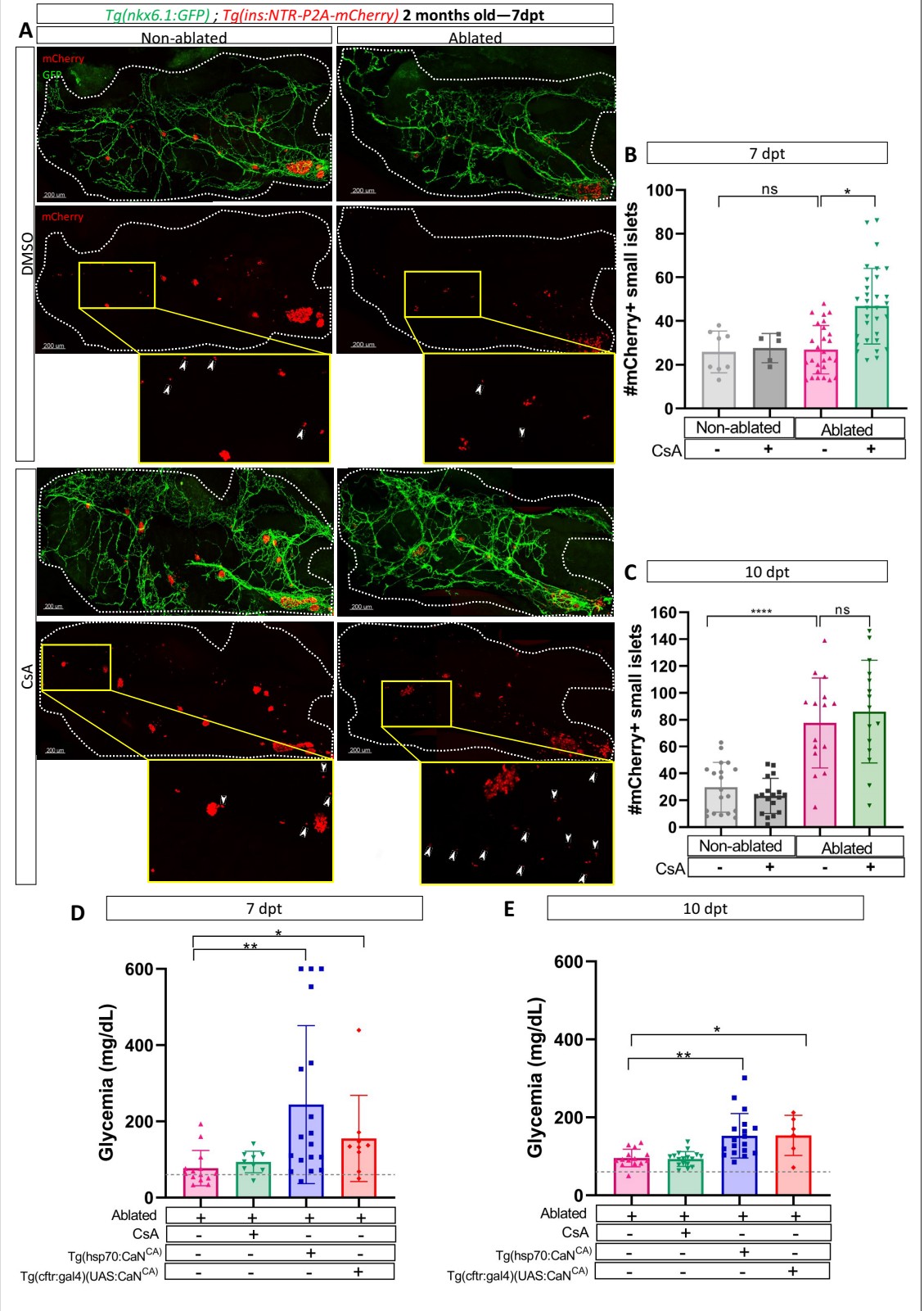

**Figure 6.** Calcineurin (CaN) regulation is important in juveniles/adults and necessary for correct glycemia recovery. (**A**) Whole mount fluorescent immunohistochemistry (GFP and mCherry) of the pancreas of *Tg(ins:NTR-P2A-mCherry); Tg(nkx6.1:GFP)* 2-month-old zebrafish at 7 dpt. 3D projection (stack) of non-ablated and ablated larvae treated with DMSO or Cyclosporin A (CsA) representative samples. The principal islet (PI) and the pancreatic tail are shown. Zoom from section of the pancreatic tail are shown, arrows point out mCherry+ beta cells in these zoom. Scale 200 μM. (**B–C**) Barplot

*Figure 6 continued on next page*

*Figure 6 continued*

representing the number of number of mCherry+ small secondary islets (≤5 cells) in the pancreatic tail of *Tg(ins:NTR-P2A-mCherry); Tg(nkx6.1:GFP)* 2-month-old zebrafish at 7 (**B**) and 10 dpt (**C**). The gray spheres represent non-ablated condition (n=8; 18); the pink triangles represent the ablated condition (n=27; 15); the black squares CsA condition (n=5; 17), and inverted green triangles ablated + CsA condition (n=30; 15). Data are presented as mean values ± standard deviation (SD). Two-way ANOVA with Tukey's multiple comparison test, *p-value <0.05, ****p-value <0.00005. The experiment was performed at least two times and the data are combined in the graph. (**D–E**) Barplot representing the glycemia (mg/dl) of *Tg(ins:NTR-P2A-mCherry)*; adult zebrafish at 7 (**D**) and 10 dpt (**E**). The pink triangles represent the ablated condition (n=13; 14); the inverted green triangles ablated + CsA condition (n=9; 17); the blue squares *Tg(hsp70:CaN^{CA})* after heat-shocks (n=17; 18); the orange lozenges *Tg(UAS:CaN^{CA}); Tg(cftr:gal4) (n=9; 6)*. The gray line represents the mean glycemia of controls (non-ablated) fish. Data are presented as mean values ± SD. One-way ANOVA with Tukey's multiple comparison test, *p-value <0.05; **p-value <0.005. Experiment performed in one technical replicate with several biological replicates (*n* visible on the graph).

The online version of this article includes the following figure supplement(s) for figure 6:

**Figure supplement 1.** Calcineurin (CaN) regulation is important in juveniles/adults and necessary for correct glycemia recovery.

Cyclosporin A (i.e. the CaN inhibitor we used in this study) as immunosuppressive drug they indeed present an increased risk of skin cancer, notably due to keratinocyte senescence inhibition (**Wu, 2010**).

It has been shown that Notch inhibitory treatments switch progenitors from proliferative self-renewing to premature differentiation, leading to progenitor depletion (**Parsons et al., 2009**; **Ninov et al., 2012**). Our study reveals that this phenomenon is further exacerbated by CaN inhibition. Importantly, during normal larval development in absence of Notch inhibitory treatment, CaN does not affect basal ductal proliferation nor beta cell differentiation. Hence, Notch signaling has to be

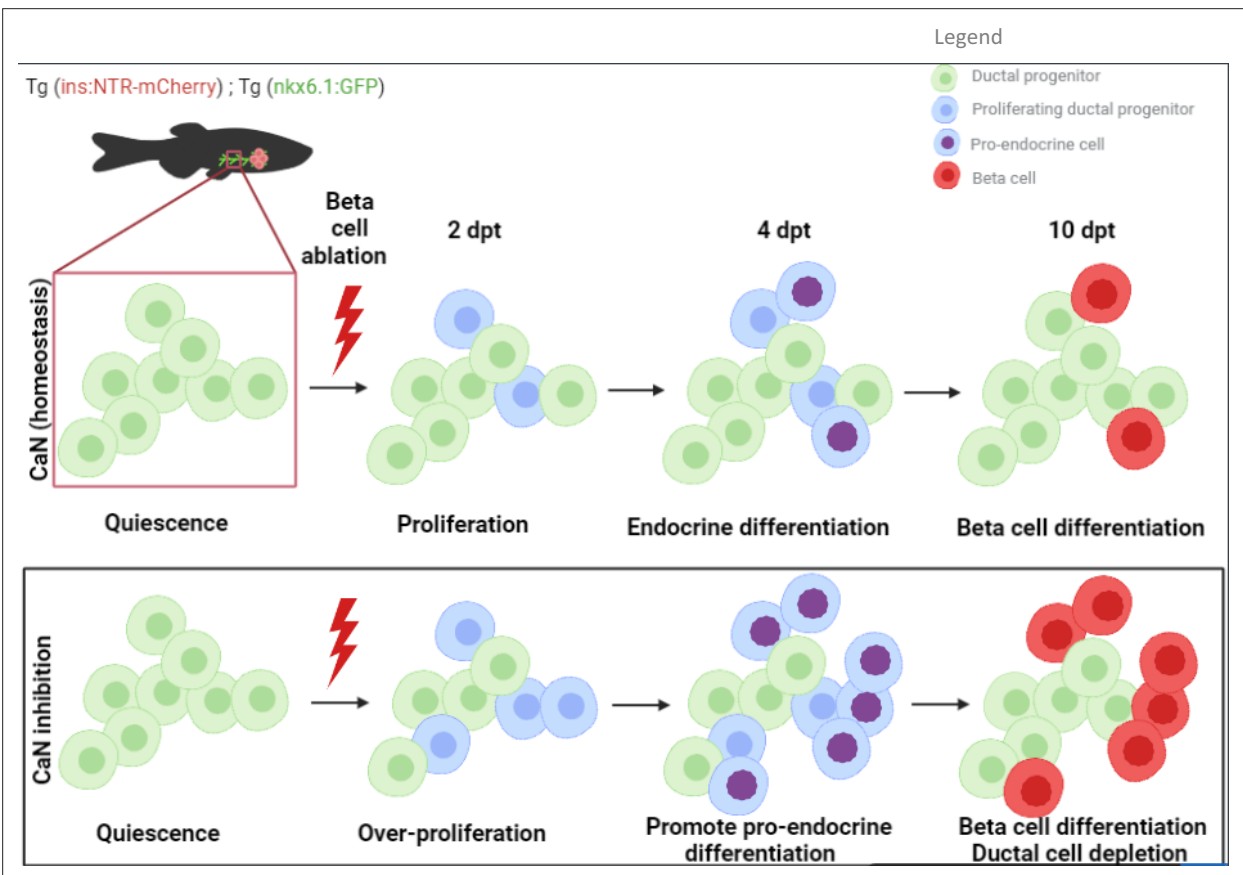

**Figure 7.** Model of calcineurin (CaN) action on ductal progenitors to regenerate beta cells. Under physiological conditions, the behavior of the ductal progenitors is determined by Notch signaling. CaN is active in these progenitors and enable a proper control between proliferation and differentiation. When CaN in repressed, more ductal progenitors enter in the cell cycle (2 dpt) and switch to a mode of proliferation leading to differentiation of the two daughter cells (4 dpt), as more pro-endocrine cells are formed. The result is a exhaustion of the progenitors and a premature beta cell differentiation (10 dpt).

repressed to detect the effect of CaN on the progenitors, suggesting that CaN acts downstream of Notch pathway. In differentiating keratinocytes, CaN cooperates with Notch signaling to regulate p21/*cdkn1a* (which is upregulated in the ducts at 3 dpt), cell cycle withdrawal and differentiation (*Mammucari et al., 2005*). These studies show that CaN acts in association with Notch signaling on progenitors proliferation and on their differentiation.

Based on our data, we build the model depicted in *Figure 7A*. Our study suggests that CaN acts in competent progenitor cells and that this competence is determined by Notch signaling. When Notch is repressed to a mild level, the progenitor enter into the cell cycle and acquire a pro-endocrinogenic competence (*Ninov et al., 2012*). CaN acts on these progenitors to tone down an excessive proliferation and avoid the exhaustion of these progenitors. CaN is therefore a guardian of the progenitor population. CaN inhibition both increases progenitor proliferation and induces their depletion, suggesting a switch to a symmetric division resulting in two daughter cells entering in endocrine differentiation.

More precisely, a previous study in mice uncover the existence of three division modes of the pancreatic progenitors during embryonic development, that is symmetric self-renewing resulting in two progenitors cells; asymmetric resulting in a progenitor and a endocrine cell and symmetric differentiative resulting in two differentiated cells (*Kim et al., 2015*). The authors actually show that the type of division is defined by the timing of induction of endocrine program by NEUROG3 (*Kim et al., 2015*). Interestingly, NEUROG3 seems to be the link between proliferation and differentiation, and its expression is regulated by Notch signaling (*Krentz et al., 2017*). In zebrafish, endocrine differentiation is not induced by *Neurog3* but by *Ascl1b* and *Neurod1* (*Flasse et al., 2013*). Concerning that subject, CaN inhibition accelerates the formation of neurod1+ cells. Hence CaN could possibly act via the determination of the type of division that is symmetric versus asymmetric. This model is supported by previous observation in others systems. In stem cells and neuronal and hematopoietic progenitors, premature differentiation results from a switch in the mode of cellular division, from symmetric amplifying division to asymmetric differentiating division (*Huttner and Kosodo, 2005*; *Ho and Wagner, 2007*). Notch determines the choice between both types of divisions (*Bultje et al., 2009*; *Guo et al., 1996*).

CaN is known to be implicated in cellular senescence (*Wu, 2010*). Usually thought as negative regulators of development and cellular growth, DNA repair (*Sousounis et al., 2020*) and cellular senescence (*Da Silva-Álvarez et al., 2020*) appear to be required in both developmental and regenerative processes. Our transcriptomic data suggest that these mechanisms are required for beta cell regeneration. Therefore, it would be interesting to determine the contribution of these cellular processes in the ductal progenitors and determine if CaN acts via cellular senescence in this case.

Overall, this study brings new insights on beta cell regeneration and highlights the ductal progenitor cell cycle as a cornerstone in the process. Some studies report an increase of proliferation of some ductal cell population in diabetic patients (*Qadir et al., 2020*; *Moin et al., 2017*), implying a regenerative response. However, these ductal cells cannot efficiently reform the beta cell mass, suggesting a dormant mechanism of regeneration. As such, the balance between proliferation and induction of endocrine differentiation could be a key to improve beta cell neogenesis. However, as CaN is also important for beta cell function, this approach would require to be transient to induce neogenesis. Therefore, it should be combined with methods to induce beta cell proliferation to ultimately reconstitute the beta cell mass. Overall, this study brings a better understanding on the regulation of the balance between ductal progenitors proliferation and endocrine differentiation. These results should provide new hints to help improve regenerative competences in mammals.

# Materials and methods

## Key resources table

| Reagent type (species) or resource | Designation | Source or reference | Identifiers | Additional information |
|---|---|---|---|---|
| Genetic reagent (*Danio rerio*) | *TgBAC(nkx6. 1:eGFP)ulg004* | PMID:26329351 | ZFIN: ZDB-ALT-160205-1 | |
| Genetic reagent (*Danio rerio*) | *Tg(ins:NTR-P2A-mCherry)ulg034* | PMID:29663654 | ZFIN: ZDB-ALT-171122-9 | |

*Continued on next page*

Continued

| Reagent type (species) or resource | Designation | Source or reference | Identifiers | Additional information |
|---|---|---|---|---|
| Genetic reagent (*Danio rerio*) | *Tg(cftr:gal4)* | PMID:25592226 | ZFIN: ZDB-FISH-150901-25442 | |
| Genetic reagent (*Danio rerio*) | *Tg(tp1:VenusPest)* | PMID:22492351 | ZFIN: ZDB-FISH-150901-8023 | |
| Genetic reagent (*Danio rerio*) | *Tg(hsp70:eGFP-P2A-ppp3ccaCA)* [ulg068] | This paper | | See Zebrafish husbandry and generation of the *Tg(hsp70:eGFP-P2A-ppp3ccaCA)* zebrafish line |
| Genetic reagent (*Danio rerio*) | *Tg(UAS:eGFP-P2A-ppp3ccaCA)* [ulg069] | This paper | | See Zebrafish husbandry and generation of the *Tg(UAS:eGFP-P2A-ppp3ccaCA)* zebrafish line |
| Antibody | Anti-GFP (chicken polyclonal) | Aves Labs | GFP-1020 | 1:1000 |
| Antibody | Anti-mCherry/dsRed (rabbit polyclonal) | Clontech | 632496 | 1:500 |
| Antibody | Anti-glucagon (mouse polyclonal) | Sigma | G2654 | 1:300 |
| Antibody | Goat polyclonal anti-Chicken IgY (H+L), Alexa Fluor 488 | Invitrogen | A-11039 | 1:750 |
| Antibody | Goat polyclonal anti-dsred 568 | Invitrogen | | 1:750 |
| Antibody | Goat polyclonal anti-Mouse IgG (H+L) Cross-Adsorbed Secondary Antibody, Alexa Fluor 633 | Invitrogen | | 1:750 |
| Chemical compound | Nifurpirinol (NFP) | Sigma-Aldrich | 32439 | |
| Chemical compound | Metronidazole (MTZ) | Sigma-Aldrich | M1547 | |
| Chemical compound | Cyclosporine A (CsA) | Selleckchem | S2286 | |
| Chemical compound | LY411575 | Sigma-Aldrich | SML0506 | |
| Chemical compound | CHIR990211 | Sellekchem | CT99021 | |
| Commercial assay or kit | Gateway LR Clonase II Enzyme mix | Invitrogen | 11791020 | |
| Commercial assay or kit | Gateway BP Clonase II Enzyme mix | Invitrogen | 11789020 | |
| Sequence-based reagent | IM369 | This paper | PCR primer | gaagaaaaccccggtcctat gtcgacgaaagagccgaaag |
| Sequence-based reagent | IM380 | This paper | PCR primer | ccttacacattcccgtcagtgc |
| Sequence-based reagent | IM371 | This paper | PCR primer | CGGCTCTTTCGTCGAC ATAGGACCGGGGT TTTCTTCCACG |
| Sequence-based reagent | O226 | This paper | PCR primer | GCCACCATGGTGAGC AAGGGCGAGGA |
| Sequence-based reagent | IM370 | This paper | PCR primer | ttattagatcttatttctgatcacctcctt |
| Sequence-based reagent | IM459 | This paper | PCR primer | cacacgaattcgccgccacc ATGGTGAGCAAG GGCGAG |

*Continued on next page*

*Continued*

| Reagent type (species) or resource | Designation | Source or reference | Identifiers | Additional information |
|---|---|---|---|---|
| Sequence-based reagent | IM460 | This paper | PCR primer | ggatcggtcgagatccttac GATCTTATTTCTGATC ACCTCCTTACG |
| Sequence-based reagent | IM457 | This paper | PCR primer | GTAAGGATCTCGAC CGATCCTG |
| Sequence-based reagent | IM458 | This paper | PCR primer | GGTGGCGGCGAATTCGTG |
| Commercial assay or kit | Nextera XT DNA Library kit | Illumina | FC-131–1024 | |
| Commercial assay or kit | Click-iT EdU Cell Proliferation Kit for Imaging, Alexa Fluor 647 dye | Invitrogen | C10340 | |
| Software, algorithm | Imaris | Bitplane (http://www.bitplane.com/imaris/imaris) | RRID:SCR_007370 | Version 9.5 |
| Software, algorithm | GraphPad Prism | GraphPad Prism (https://graphpad.com) | RRID:SCR_015807 | Version 8 |
| Software, algorithm | DESeq2 | DESeq2 (https://bioconductor.org/packages/release/bioc/html/DESeq2.html) | RRID:SCR_015687 | |
| Software, algorithm | WebGestalt | WebGestalt (http://www.webgestalt.org/) | RRID:SCR_006786 | |

## Zebrafish husbandry and generation of the Tg(hsp70:eGFP-P2A-ppp3ccaCA)[ulg068] and Tg(UAS:eGFP-P2A-ppp2ccaCA)[ulg069] zebrafish lines

*Tg BAC(nkx6.1:eGFP)[ulg004]* **Ghaye et al., 2015**; *Tg(ins:NTR-P2A-mCherry)[ulg034]* (**Bergemann et al., 2018**); *Tg(cftr:gal4)* and *Tg(tp1:VenusPest)* were used. Zebrafish were raised in standard conditions at 28°C. All experiments were carried out in compliance with the European Union and Belgian law and with the approval of the ULiège Ethical Committee for experiments with laboratory animals (approval number: 2075).

The *hsp70:GFP-P2A-ppp3ccaCA* transgene has been generated by cloning a PCR fragment containing the Gateway vector pCR8/GW/TOPO. Firstly, we amplified the full length of ppp3ccaCA with primers IM369/IM380 and amplified *GFP-P2A* with overlapping regions with IM371/O226. The overlapping PCR used the primers O226/IM380. Then to obtain a truncated *ppp3ccaCA* lacking the calmodulin biding and the autoinhibitory domain, resulting in a constantly active form of CaN, we amplified the last fragments with IM370/O226 and cloned into PCR8 vector. The promoter was assembled by LR recombination with pE5-hsp70 into pDestTol2p2A from the Tol2kit (**Kwan et al., 2007**). *Tg (hsp70:GFP-P2A-ppp3ccaCA)* fish have been generated using the Tol2-mediated transgenesis (**Kawakami, 2007**). *The Tg(UAS:GFP-P2A-ppp3ccacA)* has been generated by ligation (KLD kit, NEB) of PCR fragments *GFP-P2A-ppp3ccacA* (IM459/IM460) with the UAS sequences (IM457/IM458) in plasmid from (**Distel et al., 2009**) and then inserted into pDestTol2p2A from the Tol2kit. Final constructions has been injected with transposase into wild type (WT) AB embryos.

## Beta cell ablation

Adults fish for RNA-sequencing experiment were treated with freshly prepared metronidazole (MTZ) (Sigma M1547) at 10 mM with 0.2% DMSO in fish water. Control treatments consisted of fish water containing 0.2% DMSO. Fish were treated for 18 hr in the dark. NFP (32439, Sigma-Aldrich) stock solution was dissolved at 2.5 mM in DMSO. Beta cell ablation in *Tg(nkx6.1:eGFP); Tg(ins:NTR-P2A-mCherry)* larvae was induced by treatment with 2.5 µM NFP in E3. Control treatments consisted of E3 containing 0.2% DMSO. Larvae were treated for 18 hr in the dark.

## Drug treatments

Cyclosporine A (Selleckchem, S2286), CHIR99021, and LY411575 (Sigma-Aldrich, SML0506) stock solution were dissolved at 10 mM in DMSO. Larvae treatment were, respectively, performed at 0.1

and 5 μM in E3. Control treatments consisted of E3 containing the same amount of DMSO than drug treatment. Larvae were treated for 18 hr in the dark.

## EdU incorporation assay

Zebrafish larvae were incubated in 4 mM EdU dissolved in E3 water for 8 hr, they were then directly euthanized in tricaine and fixed in 4% PFA. EdU was detected according to the protocol of Click-iT EdU Cell Proliferation Kit for Imaging, Alexa Fluor 647 (Thermo Fisher C10340) after whole mount immunodetection.

## Heat-shock

Successive heat-shocks of 30 min and 12 hr apart were performed at 39°C for larvae and 37°C for juveniles and adults zebrafish.

## Whole mount immunodetection

Larvae were euthanized in tricaine and fixed in 4% paraformaldehyde (PFA) at 4°C for immunohistochemistry (IHC). The digestive tract of juveniles was dissected prior immunodetection and kept in methanol for at least 18 hr. After depigmentation with 3% $H_2O_2$/1% KOH, larvae were permeabilized 30 min in phosphate-buffered saline (PBS)/Triton X-100 and incubated for 2 hr in blocking buffer (4% goat serum/1% bovine serum albumin [BSA]/PBS/0.1% Triton X-100). Primary and secondary antibodies were incubated at 4°C overnight.

| | Fixation duration (hr) | Depigmentation duration (min) | Permeabilization solution | Permeabilization duration (min) |
|---|---|---|---|---|
| 5–10 dpf | 18 | 15 | PBTr 0.05% | 30 |
| 13–17 dpf | 36 | 20 | PBTr 2% | 30 |
| 2 months (digestive tract) | 18 | 15 | / | / |

Primary antibodies: Living Colors Polyclonal anti-mCherry/dsRed (rabbit, 1:500, Clontech 632496), anti-GFP (chicken, 1:1000), Secondary antibodies: Alexa Fluor-488, -568, -633 (goat, 1:750, Molecular Probes).

## Flow cytometry and Fluorescence-activated Cell Sorting (FACS)

The whole pancreas from three to four fish of Tg(nkx6.1:eGFP); Tg(ins:NTR-P2A-mCherry) adult fish (6–10 months old, males and females) were dissected, collected, and washed in Hank's Balanced Salt Solution (HBSS) without $Ca^{2+}/Mg^{2+}$. Live cell dissociation was performed in Tryple Select 1× solution (Gibco) supplemented with 100 U/ml collagenase IV (Life Technologies 17104-019) and 40 μg/ml proteinase K (Invitrogen, 25530031) for 10 min at 28°C.

The GFP+ cells were selected on FACS Aria III and sorted under purity mode and after exclusion of the doublets. The purity of the sorted cells was confirmed by epifluorescence microscopy (~95 %). Cells (about 1000–5000/fish depending on the cell type) were immediately lysed with 0.5% Triton X-100 containing 2 U/μl RNAse inhibitor and stored at −80°C.

## mRNA sequencing of FACSed cells and bioinformatic analyses

cDNAs were prepared from lysed cells according to SMART-Seq2.0 (*Picelli et al., 2014*) for low input RNA sequencing and libraries were prepared with Nextera DNA Library kit (Illumina). Independent biological replicates of each cell type sequenced using Illumina NextSeq500 and obtained ~20 million 100 bp paired-end reads. Reads were mapped and aligned to the zebrafish genome GRCz11 from Ensembl gene annotation version using STAR version 2.6.1 (*Dobin et al., 2013*). Gene expression levels were calculated with featureCounts (http://bioinf.wehi.edu.au/featureCounts/) and differential expression determined with DESeq2 (*Love et al., 2014*). Expression values are given as normalized read counts. Poorly expressed genes with mean normalized expression counts <10 were excluded from the subsequent analyses. DESeq2 uses Wald test for significance with posterior adjustment of p values ($p_{adj}$) using Benjamini and Hochberg multiple testing. The differentially expressed (DE) genes

identified with a $p_{adj}$ cutoff of 0.05 were submitted for GO analysis using WebGestalt tool (*Liao et al., 2019*).

## Confocal microscopy and image analysis

Images were acquired using Leica SP5 confocal microscope. We used ImageJ to count the cells and Imaris to do the pictures.

## Glycemia measurement

Glycemia measurement were performed as described in *Bergemann et al., 2018*.

## Acknowledgements

The authors thank the EOS (Excellence of Sciences) program from FNRS and more especially Dr. Emmanuel Dejardin for support. We also thank the GIGA technology platforms, GIGA-Genomics and GIGA-Imaging. The authors also thank Jérémie Zappia for critically reading the manuscript, Jordane Bourdouxhe and Marie Alice Dupont for their help with the constructions.

## Additional information

### Funding

| Funder | Grant reference number | Author |
|---|---|---|
| Fonds pour la Formation à la Recherche dans l'Industrie et dans l'Agriculture | | Laura Massoz<br>David Bergemann<br>Arnaud Lavergne<br>Caroline Désiront |
| Fonds De La Recherche Scientifique - FNRS | | Bernard Peers<br>Marianne M Voz<br>Isabelle Manfroid |
| Fondation Léon Fredericq | | Laura Massoz<br>David Bergemann |

The funders had no role in study design, data collection, and interpretation, or the decision to submit the work for publication.

### Author contributions

Laura Massoz, Conceptualization, Data curation, Software, Formal analysis, Investigation, Methodology, Writing - original draft, Writing – review and editing; David Bergemann, Conceptualization, Investigation; Arnaud Lavergne, Software; Célia Reynders, Caroline Désiront, Chiara Goossens, Investigation; Lydie Flasse, Writing – review and editing; Bernard Peers, Resources; Marianne M Voz, Resources, Supervision, Writing – review and editing; Isabelle Manfroid, Conceptualization, Resources, Data curation, Supervision, Funding acquisition, Methodology, Project administration

### Author ORCIDs

Laura Massoz ⬦ https://orcid.org/0000-0002-4791-0920
Isabelle Manfroid ⬦ https://orcid.org/0000-0003-3445-3764

### Ethics

This study was performed in strict accordance with the recommendations from the University of Liege ethic committee (number 2075).

Reviewer #1 (Public review): https://doi.org/10.7554/eLife.88813.4.sa1
Reviewer #2 (Public review): https://doi.org/10.7554/eLife.88813.4.sa2
Author response https://doi.org/10.7554/eLife.88813.4.sa3

## Additional files

### Supplementary files
• Supplementary file 1. Table of DE (differentially expressed) genes od ductal cells 3 days after beta cell ablation compared to non-ablated controls.
• Supplementary file 2. Table of ORA analysis results with data from *Supplementary file 1*.
• MDAR checklist

### Data availability
Sequencing data have been deposited in GEO under accession code GSE212124. The authors declare that all other data supporting the findings of this study are available within the paper and its supplementary files.

The following dataset was generated:

| Author(s) | Year | Dataset title | Dataset URL | Database and Identifier |
|---|---|---|---|---|
| Bergemann D | 2023 | Effect of beta cell ablation on pancreatic ductal cells in adult zebrafish | https://www.ncbi.nlm.nih.gov/geo/query/acc.cgi?acc=GSE212124 | NCBI Gene Expression Omnibus, GSE212124 |

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
