## [Editor Report · eLife assessment]

This work presents some **valuable** information regarding the molecular mechanisms controlling the regeneration of pancreatic beta cells following induced cell ablation in zebrafish. Specifically, the data suggest that Calcineurin is a regulator of beta cell regeneration. However, the study lacks the critical lineage tracing results to support the conclusion about the origin of the regenerated beta cells and thus is deemed **incomplete**.

---

## [Referee Report · Reviewer #1 (Public review)]

Induction of beta cell regeneration is a promising approach for the treatment of diabetes. In this study, Massoz et.al., identified calcineurin (CaN) as a new potential modulator of beta cell regeneration by using zebrafish as model. They also showed that calcineurin (CaN) works together with Notch signaling calcineurin (CaN) to promote the beta cell regeneration. Overall, the paper is well organized, and technically sound. However, some evidences seem weak to get the conclusion.

---

## [Referee Report · Reviewer #2 (Public review)]

This work started with transcriptomic profiling of ductal cells to identify the upregulation of calcineurin in the zebrafish after beta-cell ablation. By suppressing calcineurin with its chemical inhibitor cyclosporin A and expressing a constitutively active form of calcineurin ubiquitously or specifically in ductal cells, the authors found that inhibited calcineurin activity promoted beta-cell regeneration transiently while ectopic calcineurin activity hindered beta-cell regeneration in the pancreatic tail. They also showed similar effects in the basal state but only when it was within a particular permissive window of Notch activity. To further investigate the roles of calcineurin in the ductal cells, the authors demonstrated that calcineurin inhibition additionally induced the proliferation of the ductal cells in the regenerative context or under a limited level of Notch activity. Interestingly, the enhanced proliferation was followed by a depletion of ductal cells, suggesting that calcineurin inhibition would exhaust the ductal cells. Based on the data, the authors proposed a very attractive and intriguing model of the role of calcineurin in maintaining the balance of the progenitor proliferation and the endocrine differentiation. However, the conclusions of this paper are only partially supported by the data as some evidence of the lineage between ductal cells and beta cells remains suggestive.

---

## [Author Response]

The following is the authors’ response to the previous reviews.

Thank you for all your recommendations to improve the manuscript. We took them into account and tried to integrate them as much as possible in the paper. I understand that the main issue is the lack of genetic lineage tracing. Unfortunately, I am no longer in a position to perform experiments and as a consequence, we cannot bring these data. However, we previously performed several experiments that attest the ductal origin of the beta cells. As a reminder, we used experiment setting where beta cell regeneration occur from the ducts in the pancreatic tail; we used a genetic approach to over-express CaN specifically in the ducts at the level of the pancreas ; and we investigate the function of CaN under Notch repression, known to trigger beta cell formation from the ducts. Altogether, our data underline the contribution of the ductal cells. In addition, as recommended by the editors, we showed that while the proportion of ductal cells EdU+ increase Figure 5 C-D, the number of ductal cells remain constant Figure 5A supplemental. We integrate a paragraph in the discussion to remind all these points in the manuscript.

We thank you greatly for your time and consideration for this work.